# Integrating Endocrine, Genomic, and Extra-Skeletal Benefits of Vitamin D into National and Regional Clinical Guidelines

**DOI:** 10.3390/nu16223969

**Published:** 2024-11-20

**Authors:** Sunil J. Wimalawansa, Scott T. Weiss, Bruce W. Hollis

**Affiliations:** 1CardioMetabolic and Endocrine Institute, North Brunswick, NJ 08902, USA; 2Harvard Medical School, Channing Division of Network Medicine, Boston, MA 02115, USA; scott.weiss@channing.harvard.edu; 3Medical University of South Carolina, Charleston, SC 29425, USA; hollisb@musc.edu

**Keywords:** vitamin D, 25(OH)D, 1,25(OH)_2_D, epidemiology, human diseases, morbidity, mortality, prevention, treatment, public health, osteoporosis, pregnancy

## Abstract

Background/Objectives: Vitamin D is essential for bone health, immune function, and overall well-being. Numerous ecological, observational, and prospective studies, including randomized controlled clinical trials (RCTs), report an inverse association between higher serum 25-hydroxyvitamin D [25(OH)D; calcifediol] levels in various conditions, including cardiovascular disease, metabolic disorders such as diabetes and obesity, susceptibility to infection-related complications, autoimmune diseases, and all-cause mortality. Results: Vitamin D operates through two distinct systems. The endocrine system comprises the renal tubular cell-derived circulatory calcitriol, which primarily regulates calcium homeostasis and muscular functions. In contrast, intracellularly generated calcitriol in peripheral target cells is responsible for intracrine/paracrine system signaling and calcitriol–vitamin D receptor-mediated genomic effects. Government-appointed committees and health organizations have developed various clinical practice guidelines for vitamin D supplementation and management. However, these guidelines heavily relied on the 2011 Institute of Medicine (IoM) report, which focused solely on the skeletal effects of vitamin D, ignoring other body systems. Thus, they do not represent maintaining good overall health and aspects of disease prevention. Additionally, the IoM report was intended as a public health recommendation for the government and is not a clinical guideline. Discussion: New country- and regional-specific guidelines must focus on healthy nations through disease prevention and reducing healthcare costs. They should not be restricted to bone effect and must encompass all extra-skeletal benefits. Nevertheless, due to misunderstandings, medical societies and other governments have used faulty IoM report as a foundation for creating vitamin D guidelines. Consequently, they placed disproportionate emphasis on bone health while largely overlooking its benefits for other bodily systems, making current guidelines, including 2024, the Endocrine Society less applicable to the public. As a result, the utility of published guidelines has been significantly reduced for clinical practice and RCTs that designed on bone-centric are generate misleading information and remain suboptimal for public health and disease prevention. Conclusions: This review and its recommendations address the gaps in current vitamin D clinical practice guidelines and propose a framework for developing more effective, country and region-specific recommendations that capture the extra-skeletal benefits of vitamin D to prevent multiple diseases and enhance public health.

## 1. Introduction

Insufficient vitamin D is the most widespread micronutrient deficiency globally [1,2]. It is also the most misunderstood, particularly concerning the serum and tissue levels required to prevent adverse health outcomes and the daily intake necessary to maintain recommended therapeutic levels [3,4]. In the following sections, we will explore vitamin D’s biochemistry and physiological aspects and examine the reasons behind the confusion surrounding its pleiotropic effects on human health.

Vitamin D is vital for bone health, immune function, disease prevention, and overall well-being. Numerous ecological and observational prospective studies have demonstrated a strong inverse association between higher serum 25-hydroxyvitamin D [25(OH)D; calcifediol, the marker for vitamin D status] concentration and various disorders, such as cardiovascular, renal, and pulmonary disorders, metabolic conditions such as diabetes and obesity, susceptibility to infection-related complications, autoimmune diseases, and all-cause mortality. Vitamin D functions through two distinct classifications [4,5]. The endocrine system regulates calcium homeostasis, while the intracrine/paracrine and the genomic systems modulate all other systems.

The intracrine signals modulate the cells within, while paracrine signals influence adjacent cells. Meanwhile, the genomic effects are initiated following calcitriol binding to cytosolic vitamin D receptors (VDRs or, more accurately, calcitriol receptors, CTRs), which then move into the nucleus and interact with the DNA that controls most other physiological functions [5]. This review and its recommendations address the deficiencies in current vitamin D clinical practice guidelines and propose a background–framework for developing tailored recommendations that better serve clinical practice and public health needs in individual countries.

### 1.1. Importance of Different Vitamin D Metabolites and Their Biochemical Functions

Even at the expense of peripheral target cells, renal tubular cells (as well as parathyroid, fat, and muscle cells) can extract 25(OH)D from the circulation (against a concentration gradient) via the megalin–cubilin receptor system. Evolutionarily, this mechanism is designed to ensure sufficient circulatory 1,25-dihydroxyvitamin D [1,25(OH)_2_D; calcitriol] concentrations for the hormonal form of calcitriol on target cells, such as intestinal epithelia (for calcium absorption), bone cells (for resorption and calcification), and renal and parathyroid cells (regulated by parathyroid hormone) to conserve calcium. These processes are essential for maintaining blood-ionized calcium concentrations within a narrow range [6,7], critical for hundreds of enzymatic reactions, muscle contractions, and neural transmissions—a vital survival mechanism [8,9,10].

As a result, serum calcitriol levels can remain normal even in cases of moderate vitamin D deficiency (i.e., low 25(OH)D). Therefore, measuring calcitriol levels in the blood to assess vitamin D status is not recommended, as these levels are challenging to interpret and can be misleading when correlated with clinical disorders. The half-life of vitamin D is about one day, while the half-life of 25(OH)D is approximately three weeks [4]. Given this short half-life, daily vitamin D intake or regular sun exposure must maintain stable serum calcitriol levels and support the intracrine/paracrine system and genomic functions [11].

Vitamin D is secreted into breast milk, providing essential nutrition to the nursing infant [9,12]. However, 25(OH)D has a limited ability to enter breast milk due to its tight binding to the vitamin D-binding protein (VDBP), thus limiting its availability to the infant [11]. The vast majority of vitamin D activity in milk is due to vitamin D_3_ itself from physiologic and pharmacologic intakes of vitamin D [13,14], and most likely involves simple diffusion across cellular membranes.

Mammary cells express megalin and cubilin, which enable the endocytic uptake of 25(OH)D_3_-DBP, providing some entry of 25(OH)D into milk. However, compared to parent vitamin D, it is small [13,14]. This process is similar in most cells, where vitamin D entry depends on diffusion from circulation, leaving very little 25(OH)D for cellular entry. Finally, the concentration of calcitriol in circulation is approximately 900 times lower than that of 25(OH)D and vitamin D [7], making it a minor component to enter breast milk and peripheral target cells to any significant degree.

### 1.2. Biosynthesis and the Biology of Vitamin D

Once vitamin D enters a cell, it becomes hydroxylated to 25(OH)D and then to 1,25(OH)_2_D (calcitriol)—in renal tubules for hormonal regulation and within peripheral target cells for intracrine/paracrine actions and downstream genomic activations [2,5]. Unlike renal tubular cells, peripheral target cells, such as immune cells, do not secrete calcitriol into the circulation [7]. Instead, calcitriol is consumed within these cells, facilitating localized biological functions [7].

The serum concentration of 25(OH)D, with a half-life of approximately 21 days, is the only clinically measurable indicator of vitamin D status. It reflects the production of vitamin D from ultraviolet B (UVB) exposure and dietary and supplementary sources [15]. However, it does not accurately represent the amount of vitamin D stored in body tissues. Meanwhile, circulating 1,25(OH)_2_D (calcitriol) has a much shorter half-life of about 15 h and does not indicate vitamin D status or body stores. Due to the cellular entry mechanisms previously discussed, calcitriol levels in circulation do not significantly decrease until vitamin D deficiency becomes severe (below 10 ng/mL) [7].

The serum 25(OH)D concentration remains relatively stable with daily and once-weekly intake or daily sun exposure. In contrast, the 1,25(OH)_2_D concentration is constantly regulated to maintain serum-ionized calcium levels through feedback mechanisms involving parathyroid hormone, serum calcium, and phosphate. Vitamin D replacement therapy or safe sun exposure aims to normalize serum 25(OH)D levels in individuals with deficiency or insufficiency [16]. Achieving serum 25(OH)D concentrations between 30 and 60 ng/mL can alleviate symptoms, reduce the risk and severity of various metabolic disorders, and lower the incidence of falls and associated fracture risks [17,18].

The Institute of Medicine (IoM) and the latest (2024) guidelines from the Endocrine Society (TES) have adopted a “one-size-fits-all” approach, which is unsuitable for micronutrients like vitamin D. This approach oversimplifies the complexity of vitamin D, much like how a child is not simply a smaller version of an adult. Unlike pharmaceutical agents, many micronutrients, including vitamin D, are threshold compounds [19]. As a result, smaller doses may have little effect, and once a specific intake or blood concentration is reached, the benefits plateau [2,20]. Beyond this threshold, higher doses do not yield additional benefits [7]. This limitation can be addressed by calculating daily vitamin D requirements based on body weight (e.g., IU or µg per kg) rather than prescribing a universal dose for all adults [20,21]. Moreover, once a person reaches this threshold, continuing the same daily or weekly dose of vitamin D will not further increase their serum 25(OH)D concentrations.

### 1.3. Vitamin D Biological Systems

Vitamin D deficiency is the most common nutritional deficiency globally and is present in most communities in all age groups [22]. Common manifestations of vitamin D deficiency are not specific but include general ill health (not feeling well and aches and pains), proximal muscle weakness, muscle aches, low back pain, etc. Severe vitamin D deficiency (i.e., vitamin D concentration below 12 ng/mL) can present with pelvic-girdle myopathy and difficulty getting up from a sitting position, and levels below 20 ng/mL leads to loss of balance and walking impairment [23]. Vitamin D deficiency also causes immune-mediated inflammatory myopathies, aggravating muscle weakness [24,25]. Many physicians and scientists consider serum 25(OH)D levels between 20 and 30 ng/mL (50 to 75 nmol/L) as vitamin D insufficiency and less than 20 ng/mL (50 nmol/L) as a deficiency, focusing most on the muscular-skeletal tissue effect [26,27,28,29].

Vitamin D is widely known for its critical role in bone health. However, it is also essential for other body systems, including the immune, cardiovascular, pulmonary, and renal systems, as well as overall well-being. Vitamin D operates through two distinct systems: the endocrine system, which primarily regulates calcium homeostasis and musculoskeletal needs via circulating calcitriol, and the non-endocrine system, where intracellularly generated calcitriol drives intracrine/paracrine signaling and genomic functions [7,20]. These latter effects depend on tissue and cellular vitamin D levels (calcitriol), contributing to various multi-system functions.

The first system is the well-established endocrine function of maintaining calcium homeostasis, where a stable concentration of 25(OH)D is essential [11]. This system transports 25(OH)D to renal and parathyroid tissues via a megalin/cubilin transporter, relying on the vitamin D-binding protein [11]. The second system involves vitamin D’s intracrine, paracrine, and genomic actions, where the parent compound vitamin D plays a crucial role. A significant portion of circulating vitamin D is in an accessible free form, allowing it to diffuse into target cells [11], unlike 25(OH)D, which is tightly bound to the vitamin D-binding protein. Additionally, tissues such as the placenta and immune cells lining the epithelial and endothelial surfaces of organs like the gut and lungs can locally convert vitamin D into 25(OH)D and 1,25(OH)D, supporting localized biological functions [30], including the placenta [31].

## 2. Vitamin D Functions, Blood Levels, and Randomized Controlled Clinical Trials (RCTs)

Vitamin D is crucial in various physiological processes, including bone health, immune function, and overall well-being. Optimal serum levels of 25(OH)D to lower the risk of health conditions, such as cardiovascular disease, metabolic disorders, autoimmune diseases, and infection-related complications, differ for various body tissues (see Section 3.1).

Randomized controlled trials (RCTs) have been conducted to explore the efficacy of vitamin D supplementation in improving health outcomes. However, clinical trials often measure the impact of different dosages of vitamin D on achieving and maintaining adequate blood levels of 25(OH)D, providing valuable insights into the nutrient’s therapeutic potential and informing clinical guidelines for its use in disease prevention and management.

### 2.1. Less-Known Functions of Vitamin D

Vitamin D also has membrane-stabilizing effects, directly stabilizing endothelial and epithelial cell tissues and gap junctions [32]. Gibson et al. reported that vitamin D was the most effective among thousands of compounds screened, surpassing its metabolites, 25(OH)D and 1,25(OH)_2_D, in stabilizing endothelial cells [32]. This newly defined function may have contributed to vitamin D’s protection against COVID-19, which attacks endothelial tissue. Evidence suggests that maintaining serum 25(OH)D concentrations above 30 ng/mL (75 nmol/L), and preferably above 50 ng/mL (150 nmol/L), is necessary for long-term benefits. Achieving this requires regular daily vitamin D intake or sufficient sun exposure to maintain stable circulating vitamin D levels [16,21].

### 2.2. Many Conditions Need Serum 25(OH)D Above 50 ng/mL

In contrast to what is required for bone health and calcium metabolism, recent data indicate that serum 25(OH)D levels between 50 and 80 ng/mL are crucial for combating infections, cancer, and autoimmune diseases [20,21]. This is because immune cells in these conditions rely on circulating vitamin D concentrations to diffuse into the cells. In contrast, skeletal and parathyroid cells can extract lower levels of 25(OH)D from the circulation [33]. However, there is no direct way to measure tissue levels of these metabolites. In vitro studies have shown that higher serum concentrations are needed to ensure tissue sufficiency [11].

Similar serum levels have been associated with healthy pregnancy outcomes [34], reduced all-cause mortality [35], and stable vitamin D content in human milk for nursing infants [9]. Vitamin D of 1 IU of biological potency provides 0.025 µg of cholecalciferol [36]—proportionally, 40 IU = 1 µg of cholecalciferol (vitamin D_3_) equivalent of 1000 IU (=25 µg) of vitamin D_3_. However, the accuracy of the number of IUs provided cannot be guaranteed when exogenously supplemented with calcifediol [37], but the approximation is sufficient for clinical purposes. Another way of expressing this would be that supplementing vitamin D at 400 IU/day for 90 days in an adult would raise circulating 25(OH)D levels by approximately 4 ng/mL [38], but with a significant variation between individuals. If this same dose is admisntered to a newborn infant, the response is much more dramatic [39].

Achieving these serum concentrations requires a daily intake of 4000 to 10,000 IU, depending on body weight [20]. Intakes as high as 15,000 IU/day and serum 25(OH)D levels up to 125 ng/mL were reported safe with no adverse side effects [40]. However, even with the recommended daily dose of 5000 IU, it may take several months for non-obese adults with vitamin D deficiency to raise and stabilize their circulatory 25(OH)D concentrations [2,7]. Increasing these levels can be expedited with a loading dose of vitamin D [20,41,42]. In emergencies, serum levels can be rapidly raised within hours using partially activated vitamin D, such as calcifediol [20,43,44,45].

### 2.3. Vitamin D Intake Necessary to Maintain Serum 25(OH)D in Therapeutic Levels

Multiple studies confirmed that maintaining serum 25(OH)D concentrations of between 50 and 80 ng/mL facilitates relief of symptoms, reduces the risks and severity of various metabolic disorders, decreases the number of falls and associated risk of fractures [17,18], and reduces infections [20,21], sepsis [46,47], and all-cause mortality [48,49]. For the maintenance of serum 25(OH)D levels, as mentioned, most scientific organizations recommend (at a minimum) supplementing 800 IU/day for infants and, depending on the age, as much as 1000 to 2000 IU/day for children [50], and between 4000 and 10,000 IU/day for adults, based on body weight [20].

The deficit of vitamin D in an adult with serum levels less than 10 ng/mL could be over one million units [51]. In this case, a loading dose is generally required to fill the body stores, allowing an increased circulating 25(OH)D concentration. This can be achieved using vitamin D_3_ 50,000 IU daily or with a higher dose, depending on the urgency. In those with insufficiency, it can be managed with higher doses administered daily for a few days, a week, or twice a week, using 50,000 IU capsules [51,52,53]. The serum 25(OH)D concentration should be evaluated after 4 to 6 months to ensure the desired levels are reached. However, a stable daily vitamin D intake is preferable since it closely mimics normal human physiology [11] (Figure 1).

### 2.4. Calculating the Amount of Vitamin D for a Given Person

It is necessary to take the appropriate daily amounts of vitamin D to maintain 25(OH)D concentrations in the circulation at therapeutic levels—such as maintaining serum 25(OH)D concentrations above 50 ng/mL [20,21]. The dose needed to maintain such levels varies in different conditions but mainly with the body weight (BMI) [54,55]. One dose to fit all, as recommended by the IoM (2011) [56] and TES (2024) [57], does not work for the majority. Until recently, there were no practical methods for calculating the appropriate amount of vitamin D for individuals.

To address this gap, a large body of clinical research has provided tables in the journal *Nutrients* to aid in determining vitamin D supplementation needs based on various individual characteristics. Tables were derived using factors such as body mass index (BMI), body weight, or baseline serum 25(OH)D levels to calculate the required vitamin D in IU (Wimalawansa, SK, Nutrients, 2022) [20]. However, access to 25(OH)D blood level testing is often limited or unaffordable for many individuals. To offer a flexible range of vitamin D tailored to an individual’s needs, the author simplified the calculation using the following formula, enhancing its practicality. This formula is categorized based on three body weight groups [4]:I.Not obese (average wt.: BMI, <29): 70–90 IU/kg BWII.Moderately obese (BMI, 30–39): 100–130 IU/kg BWIII.Morbid obesity (BMI, over 40): 140–180 IU/kg BW.

Age, BMI, diet, sun exposure, supplement dosage, personal health status, and genotype all contribute to the variability in vitamin D serum levels, making it essential to personalize the dosing of vitamin D. For non-pregnant individuals, a dose of vitamin D of 1000–4000 IU/day will yield a serum concentration of 30–60 ng/mL. Pregnant women require higher intake in the 4000–8000 IU range to achieve these same serum levels. Serum levels of up to 100 ng/mL are safe and without side effects.

### 2.5. Avoiding Adverse Effects of Vitamin D and Obtaining Broader Benefits

Taking age-appropriate and recommended doses of vitamin D_3_ (between 4000 and 15,000 IU/day) is considered safe [58]. Daily vitamin D production following sun exposure is approximately 20,000 IU [59]. Therefore, it is unsurprising that a non-obese person’s safe upper limit of vitamin D_3_ is considered 15,000 IU/day [58,59,60]. Due to in-built safety mechanisms, excess exposure to sunlight does not cause vitamin D toxicity [20,21]. However, taking too many vitamin D supplements (e.g., more than 20,000 IU/day by a person with a normal BMI), especially in conjunction with calcium, may lead to increased serum calcium levels and should thus be avoided [7].

A recent meta-analysis of RCTs confirmed that vitamin D benefits several essential variables, such as blood pressure, lipid levels, glycemic levels, etc. [61]. It confirmed previous findings that the benefits of vitamin D are distinct in people with darker skin, lower baseline circulating 25[OH]D (<15 ng/mL), and lower BMI (<30 kg/m), and in those over 50 years, who had prolonged intake. The authors also confirmed that higher serum 25(OH)D concentrations are necessary for benefits like cardiovascular disorders in obese and older populations [61].

### 2.6. Limitations of Randomized Controlled Trials for Vitamin D Outcome Guidelines

Recent epidemiological data document the high prevalence of vitamin D inadequacy among elderly patients, especially patients with osteoporosis [62]. Low sunlight exposure, age-related decreases in cutaneous synthesis, and diets low in vitamin D contribute to the high prevalence of low vitamin D status [17,63] (Figure 1). Epidemiologic studies have reported a strong association between vitamin D deficiency and increased risks of cardiovascular events, cancers, such as colorectal and breast cancers, and autoimmune diseases, such as multiple sclerosis [64] and type 1 diabetes [65].

Well-performed observational and retrospective studies offer invaluable information concerning vitamin D’s association with chronic disease because they represent an individual’s lifestyle over time. This can never be achieved by RCTs involving vitamin D or any other nutrient. We need to heed the results of these trials and give them significant weight when setting vitamin D requirements. This concept will be discussed in detail later in this review.

Adequate vitamin D (the hormonal form of calcitriol) is vital for proper muscle functioning and the development and maintenance of the skeleton. However, evidence also suggests that vitamin D prevents several other diseases, including diabetes mellitus, hypertension, autoimmune diseases (asthma, multiple sclerosis, type 1 diabetes mellitus, rheumatoid arthritis, etc.), and certain common cancers [29,66]. Consequently, the body’s systems are unlikely to work optimally with insufficient vitamin D, and the risk of acquiring comorbidities is high.

## 3. Key Considerations for Vitamin D Supplementation

The recommended vitamin D supplementation varies for various populations, such as the general population, individuals at high risk of low vitamin D concentrations, and patients with chronic diseases [29,67]. Many observational and ecological studies and some RCTs support a causal relationship between hypovitaminosis and several chronic diseases [25,53,68,69]. Low vitamin D status is associated with hypertension [70], higher fracture rates and falls [71], obesity [72], increased all-cause mortality [73], and reduced physical performance [74]. This concept may provide information about the beneficial effects of vitamin D on the prevention and treatment of human diseases.

The dose–response relationships and health outcomes associated with serum 25(OH)D concentrations are not linear [75]. This non-linearity likely arises from the absence of direct methods to measure tissue levels of vitamin D, which are critical for assessing non-bone-related health effects. This issue is particularly relevant in RCTs, where baseline serum 25(OH)D and/or incorrect assumptions may lead to misleading statistics and conclusions.

Moreover, a significant challenge in clinical trials is that vitamin D supplementation in the control group can diminish the differences between the treatment and control groups, thereby reducing statistical power. Consequently, the correlations between oral doses or serum 25(OH)D concentrations and clinical outcomes may not be robust. Maintaining target serum 25(OH)D concentrations over time is clinically essential [76,77,78]. Studies have shown that for every 10 ng/mL (25 nmol/L) increase in 25(OH)D concentration, there is a significant risk reduction, with a hazard ratio (HR) of 0.64 (CI = 0.48–0.86) [79].

### 3.1. Diverse Population/Patient Groups Need Different Serum 25(OH)D Concentrations

Vitamin D plays a crucial role in various physiological functions, and its deficiency can lead to numerous pathophysiological diseases and disorders. While a serum 25(OH)D concentration exceeding 20 ng/mL is deemed adequate for supporting the musculoskeletal system—enhancing neuromuscular coordination and reflexes and helping to prevent falls and fractures—the vitamin’s pleiotropic effects [80] necessitate higher serum concentrations of 25(OH)D [29,81]. These higher concentrations are associated with reducing the risks of various conditions, including cardiovascular disease, hypertension, and metabolic disorders such as diabetes, insulin resistance, obesity, autoimmune diseases, and certain cancers [82,83].

The optimal serum 25(OH)D concentrations for achieving beneficial health outcomes can vary depending on the specific disease entity [84,85]. Recent data have reinforced the importance of maintaining diverse serum 25(OH)D levels to effectively counteract and reduce the risks of various diseases while minimizing complications linked to hypovitaminosis D [29,86,87]. For disorders beyond those affecting the musculoskeletal system, serum 25(OH)D concentrations should be kept above 30 ng/mL [29]. Examples of such conditions include cancer [88,89], type 2 diabetes [90,91], and all-cause mortality [86,92,93,94].

Maintaining serum 25(OH)D concentrations above 40 ng/mL can significantly reduce the risks associated with various diseases [29,95]. Evidence suggests that doubling serum 25(OH)D levels in the population—from, for example, 20 ng/mL to 40 ng/mL—could lead to not only a decreased risk of diseases but also a notable reduction in all-cause mortality, including premature deaths [96,97]. Figure 2 illustrates the varying steady-state serum 25(OH)D concentrations required to prevent or mitigate the effects of common diseases.

In addition, the thresholds for serum 25(OH)D concentrations vary across different disease states. The increased fracture risks associated with hypovitaminosis D may stem from several factors, including poor neuromuscular coordination and reflexes, which lead to a higher incidence of falls, lower calcium intake or status, other nutrient deficiencies, and secondary hyperparathyroidism. However, there is disagreement regarding the necessity of higher thresholds for specific disease categories, particularly among older individuals and those with a high body mass index [98].

### 3.2. Vitamin D Supplementation During Pregnancy

Looking at vitamin D requirements during pregnancy is one of the few areas where meaningful RCTs could be conducted, as results can be observed within nine months. A Canadian study suggested that mothers must be supplemented during pregnancy, breastfeeding, and provide supplements for infants to ensure optimal infant vitamin D status [99]. However, only a handful of RCTs on prenatal vitamin D have been conducted [100]. Early observational studies proved that vitamin D status during pregnancy could improve outcomes [100,101,102,103,104,105]. The initial vitamin D RCTs conducted in the 1980s were crude and lacked specific goals, resulting in no actionable data [106,107].

To investigate the role of meaningful vitamin D supplementation during pregnancy, Hollis et al. conducted an RCT [108]. They supplemented pregnant women from the first trimester with 4000 IU/day of vitamin D throughout their pregnancy, despite the prevailing agreement that the upper limit of vitamin D was 2000 IU/day. The project’s primary goals were to ensure the safety of the administered dose and to examine calcium metabolic factors and bone mineral density in both mothers and infants [109]. This study was the first RCT to demonstrate that vitamin D supplementation during pregnancy could improve birth outcomes, validating previous observational trials regarding the relationship between vitamin D and birth outcomes [105].

Vitamin D supplementation requirements during pregnancy have been a topic of controversy. A Canadian study suggested that to ensure optimal infant vitamin D status, both infants and mothers need supplementation during pregnancy and breastfeeding [99]. Past and recent RCT data have confirmed earlier observational study findings, emphasizing the importance of maintaining adequate vitamin D levels during pregnancy for improved pregnancy-related outcomes [100,101,102,103].

Recent data support using 4000 IU of vitamin D_3_ daily before and during pregnancy, normalizing serum 25(OH)D concentrations, reducing maternal and fetal risks, minimizing comorbidities, and improving birth outcomes [27,78,109]. Several research groups have found that maintaining maternal serum 25(OH)D concentrations above 40 ng/mL is associated with multiple beneficial clinical outcomes [78,109,110,111]. These levels can be achieved through oral vitamin D supplementation ranging from 4000 to 6000 IU per day during pregnancy [78,109,110,111,112].

### 3.3. Parenteral Nutrition

The Australian and New Zealand endocrine societies have published guidelines for providing micronutrient supplementation in adult patients receiving parenteral nutrition [113]. The guidelines highlight that vitamin D is the most vulnerable micronutrient in these populations and should be included in all parenteral nutrition programs.

Several studies indicate a high prevalence of vitamin D deficiency in patients undergoing long-term parenteral nutrition [114,115]. However, the recommended doses must be titrated to individual needs, and clinicians should exercise caution to prevent potential overdosing that could result in vitamin D toxicity [113]. Monitoring of serum 25(OH)D, parathyroid hormone, calcium, magnesium, and phosphate concentrations is advised, with appropriate dose adjustments to minimize the risk of developing metabolic bone disease [116].

### 3.4. Effective Food Fortification Strategies Needed to Alleviate Vitamin D Deficiency

Food fortification is a proven strategy to alleviate micronutrient malnutrition [117]. It is a cost-effective method for nutrient delivery, reaching a broad population, as only highly motivated individuals consistently adhere to long-term supplementation. However, implementing food fortification requires navigating through government agencies, which can be challenging.

Research on compliance with food-based dietary guidelines and increased vitamin D fortification has been conducted in several countries, including the United Kingdom (*n* = 911), the Netherlands (*n* = 1526), and Sweden (*n* = 974) [118]. The findings indicate that doubling vitamin D levels in products such as margarine and milk increased vitamin D intake by approximately 40% (ranging from 4.0 to 10 mcg/day), with minimal variations observed between the countries [119].

Several researchers and groups have advocated for adequate levels of food fortification with vitamin D due to its numerous benefits [119,120,121,122,123]. This is especially crucial during pregnancy and lactation, with recommended doses ranging from 4000 to 6000 IU/day [34,78]. Establishing a “globally applicable guideline” for vitamin D and food fortification is timely and necessary to address the issue of over 3.5 billion people worldwide who are deficient or insufficient in vitamin D.

Rectifying vitamin D deficiency costs less than 0.1% of the expenses associated with investigating and treating comorbidities and complications linked to hypovitaminosis D, with estimates ranging from 0.06% to 0.2% [124]. On an annual basis, vitamin D supplementation costs approximately USD 12 per person [125], while managing complications related to vitamin D deficiency—including premature deaths—ranges from USD 6000 to USD 18,000 per affected individual each year [126]. Despite the significant cost–benefit advantages, millions suffer from disorders associated with vitamin D deficiency. The global (regional and country-specific) vitamin D recommendations should emphasize this considerable discrepancy and the cost-effectiveness of vitamin D supplementation [125].

### 3.5. Challenges Associated with Vitamin D Assays

Despite international efforts to standardize the laboratory measurement of 25(OH)D since 2010, there remains a lack of uniformity worldwide. Data analysis from the Vitamin D External Quality Assessment Scheme (DEQAS) across ten studies has identified potential biases stemming from the absence of assay standardization, which is necessary for accurately calibrating 25(OH)D levels [127]. The authors highlighted the challenges in achieving assay standardization and emphasized the importance of participation in a validated quality assurance program [127].

Standards for vitamin D assays can be obtained from the National Institute of Standards and Technology (NIST) for calibration purposes. Properly calibrated liquid chromatography–mass spectrometry (LC/MS) assays for 25(OH)D are considered the gold standard; however, many poorly validated LC/MS assays yield inferior results, necessitating caution. Additionally, direct 25(OH)D assays utilizing CLIA technology should never be used on cord blood samples, as the values obtained may overestimate actual concentrations by up to 50% due to interference from nonspecific matrix effects of cord blood on the direct assay [128].

Several blood-spot vitamin D testing kits are now available. These include DBS/pocket LFIA from ImmunoCeutica, ZRT Laboratory (www.zrtlab.com) and Omega Quant (www.omegaquant.com) blood-spot vitamin D testing kits used by GrassrootsHealth, Encinitas, CA, USA. These blood spot assays were validated against the liquid chromatography/tandem mass spectrometry (LC-MS/MS) for 25(OH)D. Some of these laboratories are certified by the Clinical Laboratory Improvement Amendments (CLIA) and participate in DEQAS, the Vitamin D Quality Assessment Scheme.

## 4. Key Guidelines on Vitamin D

Several scientific societies and health authorities have reviewed the literature and published their interpretations and guidelines for vitamin D supplementation for the public. The recommended minimum serum 25(OH)D target level among most guidelines is 75 nmol/L (30 ng/mL), slightly extending the scope beyond skeletal benefits to include metabolic disorders [50]. However, some guidelines, such as those from the IoM [129] and certain European societies [130,131], propose a minimum serum 25(OH)D concentration of 20 ng/mL (50 nmol/L); this is based solely on bone health [56,132]. These conclusions are based on RCTs that studied vitamin D’s endocrine effects on bone, often with very low levels of supplementation, while overlooking its other crucial roles. Contrary to the scientific literature, these recommendations fail to account for vitamin D’s significant intracrine/paracrine functions, such as signaling in Th1 macrophages and lymphocytes [7].

Nevertheless, most research confirms that the general population in many countries fails to meet the minimum recommended vitamin D levels, let alone the optimal levels. This poses a significant public health concern, as low vitamin D status increases the risk of various health issues across all age groups and genders [133,134]. The critical public health and patient-oriented guidelines addressing this issue are discussed later.

Government-appointed committees and health organizations have developed various clinical practice guidelines for vitamin D supplementation and management. However, these guidelines have heavily relied on the 2011 Institute of Medicine (IoM) report, which primarily focused on the skeletal effects of vitamin D [56]. Although the IoM report was a public health recommendation rather than a clinical guideline, subsequent vitamin D guidelines have used it as their foundation, focusing predominantly on bone health and overlooking its broader benefits for other bodily systems. Consequently, the utility of these guidelines has diminished significantly for clinical practice and remains suboptimal even for public health and disease prevention.

Despite numerous clinical trials showing significant non-skeletal benefits of various vitamin D doses without adverse effects, government agencies such as the US Preventive Services Task Force [135], the National Institutes of Health (USA), the Scientific Advisory Committee (SCAN), the National Institute for Health and Care Excellence (NICE) [53,77,136] and the National Osteoporosis Society [137] in the UK continue to assert that there is no compelling evidence supporting vitamin D supplementation, including for older men and premenopausal women in community settings.

However, these conclusions were derived solely from selective analyses of RCTs that included certain studies while excluding positive findings. Consequently, despite claims, they reflect the subjective opinions of appointed committee members [135,137] rather than the full spectrum of published science. This underscores the need for unbiased selection and balanced consideration of better-designed RCTs and using them in meta-analyses to generate accurate and meaningful conclusions.

### 4.1. Institute of Medicine Report (IoM) [56]

The IoM Report 2011 was not a research study, but a summary of “opinions” based on a ‘selected’ small number of published datasets reviewed by experts [56]. It is important to note that the IoM guideline does not apply to patients or those who live outside North America [138,139]. Moreover, this report was criticized for not including relevant trials and data and a significant statistical error in their calculations of the recommended dose and serum 25(OH)D levels [140,141,142,143].

The RDA and the IoM panel concluded that 20 ng/mL (50 nmol/L) was the correct goal for vitamin D blood levels [25(OH)D] [56]. However, it only focuses on bone health and neglects other body systems. Thus, it cannot be considered a logical or physical minimum level. The RDA was set to allow 97.5% of the population to reach the desired level, but the set level is only sufficient for the musculoskeletal system. Only selecting ten RCTs exclusively on bone health neglected thousands of other clinical studies that reported on other body systems.

Thus, the foundation of the IoM report is flawed and, thus, cannot be considered a logical or physical minimum level (see Section 4.1.2). Despite the IoM suggestion that 20 ng/mL of serum 25(OH)D is adequate for all humans [144], data from most cross-sectional and ecological studies and many RCTs strongly contradict this assertion [11,52,53,69,145]. Most studies report that the optimal serum 25(OH)D concentration for the pleiotropic effects of vitamin D is at least 30 ng/mL or greater [50,52,67] and disagree with the IoM recommendations [129,146,147,148].

The IoM was tasked with developing a “public health guideline” for North America. It was not designed as a clinical guideline for assessing vitamin D adequacy or for use in clinical settings outside North America [139]. Despite published clinical trial data being available to firm conclusions, the IoM committee admitted that further research was needed to determine a definitive daily recommended intake [56]. Some scientific societies and countries misunderstood the IoM report. While the report may serve as a public health resource for the US government, its recommendations are not intended for other uses, such as clinical trials, clinical practice, or applications outside North America.

#### 4.1.1. Intentions of the IoM

Despite these caveats, many organizations have unfortunately misinterpreted the IOM’s conclusions. The IoM guidance [56] pertains solely to the “skeletal effects” of vitamin D, aiming to prevent rickets [149] in children and osteomalacia in adults [150]. This forms the basis for its recommendation of 20 ng/mL (50 nmol/L) as an adequate serum 25(OH)D concentration [50,52,67,151]. Consequently, these guidelines do not apply to disease states, clinical practice, or populations outside North America [53,147,152]. Moreover, the IoM’s bone-centric recommendation of a 25(OH)D level of 20 ng/mL (50 nmol/L) corresponds to a daily vitamin D intake of only 400 to 800 IU, excluding the effects of UVB exposure [141], which contributes less than 5% of vitamin D production [153,154]. In contrast, 30 min of safe sun exposure to a fair-skinned person can generate between 10,000 and 20,000 IU of vitamin D daily without causing sunburn [59].

After reviewing IoM data, independent authors discovered significant statistical errors in the IoM report that underestimated the vitamin D intake [140], further weakening its validity and utility [155]. The recommended intakes assessed were based on achieving a 25(OH)D of 20 ng/mL on average and not on 97.5% of people, which should have been the aim. In addition, many people, including the obese, those with gastrointestinal disorders, and those who are taking medications that increase the catabolism of vitamin D, need to take eight to ten times greater doses than those advised by the IoM [69,75,94]. Notably, the IoM guidelines do not apply to patients.

In brief, the IoM guidelines were not designed for individual patients, especially those at high risk for vitamin D deficiency [56], or for those with conditions such as cardiovascular disease, metabolic syndrome, cancer, insulin-dependent diabetes, type 2 diabetes (T2D), asthma, and multiple sclerosis [64]. Individuals with these disorders require higher steady-state serum 25(OH)D levels, typically over 50 ng/mL, to manage their conditions effectively [51,87,95]. Furthermore, research from various groups supports that the serum 25(OH)D concentrations needed to reduce disease risks can vary significantly [86,87].

#### 4.1.2. Why the IoM Vitamin D Intake Recommendations Are Invalid

The ten clinical studies used by the IoM for RDA calculations included multiple dosing groups from 32 individual studies [156]. The fitted dose–response curve (green solid line in Figure 3A below) represents the best fit through the averages [156]. The confidence interval lines (green dashed lines) depict the upper and lower bounds within which the best-fit line falls with 95% certainty. The lower 95% confidence interval corresponds to the 2.5th percentile, meaning that 97.5% of the population would be expected to achieve a serum level above this value [156].

Veugelers and Ekwaru’s work (Figure 3B below) illustrates the IoM [56] chart to calculate the RDA as 600 IU/day. Based on these data, the IoM estimated that 600 IU/day would bring the average individual to a serum 25(OH)D level of 25 ng/mL (63 nmol/L) and 97.5% of the population to 22.5 ng/mL (56 nmol/L). For added caution, they rounded down to estimate that 600 IU/day should elevate serum 25(OH)D levels to 20 ng/mL (50 nmol/L) in 97.5% of the population, which became the RDA target [139,156].

The IoM authors calculated the averages of the 23 studies at the 2.5th percentile by subtracting two standard deviations from the mean, shown as yellow dots in Figure 3B [156]. Veugelers and Ekwaru, through regression analysis of these 23 data points, demonstrated a significantly lower prediction limit, represented by the red line in Figure 3B. Their regression indicated that with a daily intake of 600 IU of vitamin D, 97.5% of individuals would only reach serum 25(OH)D values above 26.8 nmol/L rather than the 50 nmol/L reported by the IoM [139].

Additionally, they estimated that a daily intake of 8895 IU of vitamin D is necessary for 97.5% of individuals to achieve serum 25(OH)D levels above 50 nmol/L (20 ng/mL), indicating that the required dose is over ten times higher than the current IoM recommendation. The IoM recommended daily amounts (RDAs) of 600 IU/day [139], which may not even be sufficient to maintain bone health. Therefore, the IoM’s 600 IU/day RDA and tolerable upper intake levels are substantially underestimated and thus worthless [157].

#### 4.1.3. Major Statistical Calculation Errors in IoM Report

Additionally, the GrassrootsHealth.org scientist panel contested the IoM’s RDA calculations. Instead of revisiting the same dataset, they utilized the D*action dataset, which comprised 3657 individuals who reported taking between 0 and 10,000 IU/day of vitamin D. This group provided health outcomes and supplement dosage data within recommended guidelines [158]. The panel plotted each individual’s attained serum 25(OH)D levels against their reported daily supplement intake and created 95% confidence intervals. As shown in Figure 4, the lower 97.5% confidence interval (the red line) crossed the 20 ng/mL (50 nmol/L) serum level at approximately 3875 IU/day of vitamin D supplementation.

However, humans also acquire some vitamin D from dietary sources and sunlight. According to responses from the D*-action questionnaires and data from another GrassrootsHealth paper, participants were estimated to obtain approximately 3300 IU/day from food and sunlight combined [75]. Based on these findings, the total daily vitamin D intake required to ensure that 97.5% of the population achieves serum 25(OH)D levels above 40 ng/mL was calculated to be around 7000 IU/day [75].

### 4.2. The Endocrine Society Guidelines (2011) [50]

While the IoM report [56] and the RDA [139] were public health guidance to the government, the Endocrine Society (TES) guidelines in 2021 [50] were intended for patients but not for any high-risk groups or disease prevention [50]. TES guidelines highlight that vitamin D deficiency is common in all age groups and that few foods contain helpful amounts of vitamin D. They recommend daily intake/supplementation levels and tolerable upper limits, depending on age and certain clinical circumstances [50]. Furthermore, TES suggested measuring serum 25(OH)D concentrations using newer, more reliable assays for diagnosing vitamin D deficiency [50].

TES guidelines suggest that infants (from birth to 1 year) require at least 400 IU/day (IU = 10 μg) of vitamin D, and children 1 year and older require at least 600 IU/day (15 μg) to maximize bone health [49]. For adults, at least 1000 IU/day of vitamin D is recommended to raise and maintain serum 25(OH)D concentration above 30 ng/mL (75 nmol/L). TES recommendations were made for the public, patients, and multiple ethnic groups, and are thus applicable to the general population [50]. The IoM and AES suggested a safe upper limit of 4000 IU/day of vitamin D for the population [50,56].

### 4.3. The Endocrine Society (TES) Recommendations (2024) [57]

The 2011 TES guidelines were a step in the right direction [50], but these were reversed by the newly published guidelines in 2024 [57]. Both reports primarily focus on the vitamin D needs of the skeletal system in healthy individuals, disregarding vulnerable populations, disease conditions, and sick patients. Despite focusing on the same skeletal aspect, the recommendations between the two reports from TES were contradictory. With such uncertainty, 2024 TES guidelines should not be used for designing clinical studies, disease prevention or treating individuals with vitamin D deficiency or other disorders, policy-making, or clinical practice.

Notably, the 2024 guidelines [57] disregarded thousands of new studies published since 2011 that highlight the extra-skeletal benefits of vitamin D. Although a vast amount of new data supports the necessity of higher vitamin D intake and elevated serum 25(OH)D concentrations to manage various conditions such as infections [2,159,160], cancer [161,162], autoimmune diseases [4,7,163,164], and more, TES guidelines in 2024 [57] completely ignored them.

#### 4.3.1. Major Fallacies in Endocrine Society Guidelines, 2024 [57]

Additionally, the TES (2024) guidelines recommend the same vitamin D dosage for individuals aged 1 to 75 [57], including pregnant individuals and those at high risk for pre-diabetes, suggesting a serum 25(OH)D level of 30 ng/mL as adequate [165]. However, these guidelines fail to provide therapeutic levels for high-risk populations or specific indications, rendering their recommendations ambiguous and impractical. Despite claims of an “absence of supportive clinical trial evidence”, there exists a substantial body of over a thousand well-controlled clinical studies, including many RCTs, demonstrating that the recommended serum 25(OH)D levels are insufficient for addressing most non-skeletal disorders [11,29].

The 2024 report failed to acknowledge the need for higher doses of vitamin D for optimal health, particularly in vulnerable groups and those with specific disorders [57]. These include pregnant individuals, breastfed infants, those with obesity, intestinal malabsorption disorders, and individuals who have undergone bariatric surgery. Furthermore, people with darker skin who have limited sunlight exposure, as well as housebound or indoor workers [16]—such as night-shift employees, nursing home residents, disability center occupants, prisoners, and astronauts—are at increased risk [166]. Older adults, smokers, asthmatics, and individuals with metabolic disorders (including obesity and diabetes), infections and sepsis, and autoimmune disorders (like multiple sclerosis and inflammatory bowel diseases) also require special attention regarding adequate vitamin D supplementation.

The 2024 TES recommendations [57] are limited to bone health, relying heavily on the IoM report [56]. In addition, it failed to address the needs of most of the global population and their needs. As a result, their recommendations are unsuitable for many individuals and should not be integrated into country- or region-specific guidelines or for any policy-making. A recent systematic review supports this conclusion [166]. Therefore, neither clinical practice nor clinical trial designs should reference the IoM report [56] or TES reports (2011 and 2024) [50,57] for determining doses in the clinical management of patients or interventional groups in clinical trials, national guidelines, or policy-making.

#### 4.3.2. Discrepancies Between 2024 [57] and 2011 [50] Endocrine Society Guidelines

Despite evidence showing that oral supplements for those with vitamin D deficiency can reduce the risk of various diseases and mitigate complications, the TES panel recommended against testing for 25(OH)D levels [167]. However, they did not specify the criteria for such recommendations. Overall, the 2024 guidelines undermine the 2011 TES recommendations [50], making a retrospective attempt to reaffirm the IoM report [56]. TES 2024 recommended daily intakes of 600 to 800 IU (15 to 20 µg; the doses given to infants) as sufficient for adults—doses that are only adequate for infants and preventing rickets and osteomalacia in adults. Furthermore, for individuals aged 50 to 74 years, the panel suggested against routine vitamin D supplementation, disregarding numerous published public health recommendations and studies focused on disease prevention.

The 2024 guidelines recommend that most adults take no more than the recommended daily allowance (RDA) established by the outdated 2011 IoM report, which includes vitamin D intakes of 400 IU per day for infants, 600 IU/day for all non-pregnant and pregnant adults, and 800 IU/day for adults over 70 years of age [56]. Additionally, these guidelines suggest testing 25(OH)D levels across all populations, including vulnerable groups, but fail to define a sufficient level [167].

### 4.4. European Guidelines

Compared with the vitamin D TES guidelines from North America in 2011 [50] and those in other parts of the world, the TED 2024 guidelines [167] and European recommendations have consistently been overly cautious [168,169]. Like the IoM, the European focus has primarily been on requirements for skeletal tissue and bone mineralization, disregarding extra-skeletal benefits. Consequently, the significant pleiotropic benefits in extra-skeletal tissues are often downplayed or overlooked despite substantial evidence supporting their importance [80].

European researchers concluded that the evidence regarding the associations between 25(OH)D levels and extra-skeletal chronic diseases was inconclusive, largely based on the results of RCTs [169]. This conclusion is partly due to the inclusion of poorly designed RCTs in their analysis. Consequently, European guidelines advocate for more well-structured RCTs to establish that maintaining 25(OH)D concentrations within a specific range is beneficial and safe for preventing and treating various diseases.

In contrast, the guidelines for Central European countries aim to enhance the vitamin D status of neonates, children, adolescents, adults, and seniors while serving as a resource for healthcare professionals and regulatory bodies. These Central European Guidelines provide a consensus on vitamin D supplementation and outline population strategies to eliminate vitamin D deficiency within the general population and those with hypovitaminosis D [53,170].

#### 4.4.1. Polish Guidelines

The European vitamin D supplementation guidelines published by Pludowski et al. (2018) encompass the research carried out during the past two decades, mainly focusing on the pleiotropic actions of vitamin D [53,80]. These recommendations are practical, realistic, and thus applicable [53]. The authors concluded that most observational and ecological studies demonstrated correlations between serum 25(OH)D concentrations and improved outcomes in many chronic communicable and non-communicable diseases [53].

Notably, the doses recommended for pregnancy and seniors and high-risk groups (developing severe vitamin D deficiency and having comorbidities) [171] are insufficient. They will not raise their serum 25(OH)D concentrations to needed therapeutic concentrations. In addition, the report also recommended a one-size-fits-all principle: to take 4000 IU/day or 7000 IU/week for ages between 1 and 90 years, which makes little sense. Finally, the report recommends using calcifediol, which is 15 times more expensive than cholecalciferol and is likely to cause more adverse effects [7].

The updated Polish vitamin D guidelines 2024 [171] are primarily unchanged from the original reports in 2009 [172] and 2013 [170]. They failed to incorporate the latest research findings over the past decade. They maintain the same 25-hydroxyvitamin D concentration—indicating vitamin D deficiency [<20 ng/mL (<50 nmol/L)], suboptimal status [20–30 ng/mL (50–75 nmol/L)], and optimal concentration [30–50 ng/mL (75–125 nmol/L)] [171]. The intake recommendations are like those in the IoM report.

#### 4.4.2. Italian Guidelines

The Italian Society for Osteoporosis guidelines also focus on defining, preventing, and treating vitamin D deficiency. They recommend a daily allowance ranging from 1500 to 2300 IU [173]. Consistent with other scientific societies, the Italian Society for Osteoporosis defines serum 25(OH)D concentrations below 20 ng/mL as deficient and those between 20 and 30 ng/mL as insufficient. Additionally, they emphasize that 50% of healthy young individuals are likely to have insufficient vitamin D levels [173].

In contrast to the guidelines from the United Kingdom and a few other countries, the Italian Society for Osteoporosis recommends a cumulative dose of between 300,000 and 1,000,000 IU administered for 1 to 4 weeks, followed by a maintenance dose of between 800 and 2000 IU per day. They also suggest checking serum levels once every two years [173]. The Italian Societies aligns with the TES recommendation that the daily tolerable upper intake is 4000 IU daily. Italians Medical Agencies published guidelines, Nota 96 [174], AIFA, and the Note 96 Update [175]. The update was based on the IoM and influenced by an extension of the American VITAL study (2022) [176] and the European study DO-HEALTH (2020) [177] (Bischoff-Ferrari HA et al., JAMA 2020).

The latter two studies and the primary VITAL study (all having flawed study clinical study designs and thus unreliable conclusions) concluded that long-term vitamin D supplementation with more than 2000 IU/daily would not modify the fracture risk for osteoporosis in healthy populations. A subsequent Consensus Statement on Vitamin D Status Assessment and Supplementation in 2024 [178] did not rectify or mention fundamental errors in those flawed RCTs nor add anything new to the existing knowledge or vitamin D guidelines. Despite having hundreds of peer-reviewed published clinical studies, as with most systematic reviews and meta-analyses, it continued with the jargon, “further studies are needed to investigate vitamin D effects about the different recommended 25(OH)D levels and supplements”.

#### 4.4.3. Croatian Guidelines

An evidence-based clinical guideline from Croatia was developed to detect vitamin D deficiency and guide prevention and therapy in healthy populations and patient groups [179]. The guideline recommends maintaining serum 25(OH)D levels between 30 and 50 ng/mL (75 and 125 nmol/L) and administering preventive and therapeutic dosages of vitamin D to achieve these levels [179]. Despite such guidance, tropical countries like India, Sri Lanka, and the Middle East experience a high prevalence of vitamin D deficiency, with rates exceeding 70% in healthy children and a similarly high prevalence among adults [180,181,182].

#### 4.4.4. Persian Gulf-Region Guidelines

The United Arab Emirates and the Persian Gulf region experience a high prevalence of vitamin D deficiency throughout the year. Therefore, the clinical practice guidelines for vitamin D in this region should consider the area’s unique climatic conditions, cultural habits, and lifestyle factors. These include dietary practices, insufficient physical activity, sun-avoidant behaviors (partly due to extreme heat), and other risk factors contributing to vitamin D deficiency [77]. It was emphasized that the goal of supplementation of the Persian Gulf-region guidelines is to achieve long-term maintenance serum 25(OH)D concentrations between 30 and 50 ng/mL [77]. A summary of these regional recommendations is provided in Table 1.

### 4.5. American Academy of Developmental Medicine and Dentistry (AADMD)

Guidelines for individuals with intellectual and developmental disabilities (IDDs) are available. These include the Canadian consensus guidelines for adults with developmental disabilities [183], the consensus statement from the American Academy of Developmental Medicine and Dentistry [52,68], and the European guidelines [53]. Research indicates that optimal vitamin D levels in persons with IDDs may help reduce the risk of depression [184], cognitive decline [185], and hypertension [186]. To maintain optimal serum 25(OH)D concentrations, daily supplementation of 2000 to 5000 IU of vitamin D is recommended. Alternative recommended regimens were 15,000 to 30,000 IU per week or 50,000 IU twice a month [51,187,188].

### 4.6. Guidelines for Vitamin D Supplementation Post-Bariatric Surgery

The rising prevalence of obesity has made bariatric surgery a standard surgical option for weight reduction in obese individuals and those with type 2 diabetes who require high doses of insulin to manage their condition and prevent future complications [189]. However, these procedures often result in multiple nutrient deficiencies, including micronutrients, minerals, and vitamins. These deficiencies are primarily due to the structural changes made to the gastrointestinal tract, disruptions in enzymatic function, and impaired fat absorption.

Fundamental clinical practice guidelines for this group of individuals are available from several organizations, including the TES (2011), the American Association of Clinical Endocrinologists (2013), The Obesity Society (2014), the American Society for Metabolic & Bariatric Surgery (2013), and the Interdisciplinary European Guidelines on Metabolic and Bariatric Surgery (2014). However, recommendations in these guidelines were limited partly due to a lack of statistical power and an imprecise description of the target population, which weakens their applicability in clinical settings.

A summary of these international guidelines suggests supplementing with high doses of vitamin D, ranging from 3000 IU daily to 50,000 IU one to three times per week, to address nutrient deficiencies after bariatric surgery. Due to the absence of well-designed RCTs and clinical outcome data, these recommendations are based primarily on anecdotal evidence [190]. Nonetheless, all published studies conclude that individuals undergoing bariatric surgery must comprehensively evaluate their nutrient needs and receive appropriate long-term treatment with relevant micronutrients before and after the procedure.

### 4.7. General Issues with Vitamin D Clinical Guidelines

Most guidelines (except IoM and TES 2024) emphasize the pleiotropic effects of vitamin D and recommend a minimum target serum 25(OH)D concentration of 30 ng/mL (75 nmol/L), with an optimal range between 30 and 60 ng/mL. Vitamin D doses ranging from 2000 to 4000 IU/day are typically needed to achieve the mentioned serum level, depending on baseline 25(OH)D concentrations, age, body weight, disease status, and ethnicity [53]. Many individuals, particularly those with specific conditions, may require more than 4000 IU/day to maintain serum 25(OH)D levels above the minimum recommended threshold [11,27,53,69,191]. Consequently, many scientists and clinicians worldwide agree that a serum 25(OH)D concentration of at least 30 ng/mL is essential for reducing the risk of broader diseases.

The oral dose of vitamin D required to correct the deficiency is influenced by various factors, including ethnicity or skin color, use of certain medications, and comorbidities such as obesity or malabsorption syndromes [192,193,194,195,196,197,198]. In patients with advanced chronic kidney disease, maintaining serum 25(OH)D levels above 40 ng/mL is particularly beneficial for mitigating the adverse effects of secondary hyperparathyroidism [51,53,87].

### 4.8. Misapplication of Flawed Guidelines in Healthcare Policies

Some organizations have misunderstood and misinterpreted public health guidelines as clinical practice guidelines. This issue has been observed in certain European and Eastern countries, where public health recommendations have been inappropriately incorporated into clinical guidelines. A prominent example is the National Osteoporosis Society (NOS) in the United Kingdom (UK) [199], whose statements were later adopted verbatim by the National Institute for Health and Care Excellence (NICE), a UK national guideline-setting body [200]. These guidelines were based on misinterpretations of the 2011 IoM report [56] and other public health documents without fully considering the clinical context.

The misapplication of the bone-centric guidelines, the mixing of vitamin D with other nutrients, and the inclusion of non-clinical material in national recommendations make them confusing and difficult to follow. Additionally, as mentioned above, the bone-health-driven flawed guidance of the guidelines claims that there is no evidence of benefits in other body systems and that more RCTs are needed, which is false and misleading. Consequently, if the flawed recommended daily allowance (RDA) for vitamin D and inadequate serum 25(OH)D concentration levels is applied in public health policies, it would lead to an increase in preventable ailments such as cardiovascular diseases, infections, and autoimmune disorders. Consequently, these imperfect policies would contribute to a greater disease burden, more hospitalizations, premature deaths, and higher healthcare costs.

Public health recommendations should be focused on vulnerable populations—such as older people, pregnant women, persons with dark skin living in the tropics, and individuals with limited sun exposure or sun avoidance—like routine night-shift workers in residential facilities, prisons, submarines, etc. Despite its importance, no specific recommendations have been issued by any of the leading healthcare agencies related to public health.

## 5. Country-Specific Vitamin D Clinical Practice Guidelines, Incorporating Extra-Skeletal Benefits

Country-specific vitamin D clinical practice guidelines for chronic disease prevention vary widely based on local and regional health policies, sunlight exposure, dietary habits, and public health priorities. Most of these guidelines have been influenced by the flawed Institute of Medicine (IoM) report [56] from 2011 in the USA (more information below).

### 5.1. Comparisons of Country-Specific Vitamin D Guidelines

National health authorities or medical organizations develop these guidelines to address the unique needs of their populations. However, they often lack coordination and do not adequately focus on the specific needs of the country or region. As a result, contradictory guidelines can exist within the same country, as observed in the USA. Table 2 summarizes and provides an overview of nine vitamin D guidelines to prevent chronic diseases.

### 5.2. Analyses of Country-Specific Vitamin D Guidelines

All guidelines mentioned in Table 1 were focused on bone health—preventing rickets and osteomalacia. Consequently, none of them apply to non-skeletal diseases and disorders. All guidelines mentioned in Table 1 were copied from the faulty IoM guidelines published as public health guidance for white Caucasians in North America. It is not for use in clinical practices or similar decision-making processes. While they were not intended as clinical guidelines, most countries used them based on clinical decision-making due to a misunderstanding of the IoM publication.

Rather than replicating a flawed IoM report, each country should have developed its guidelines based on regional factors, including sunlight exposure, diet, and public health concerns, independent of the USA recommendations. However, in recent years, with over 7500 peer-reviewed studies supporting higher intakes and minimum serum 25(OH)D levels, there is a growing trend across many countries to consider the broader role of vitamin D in preventing chronic diseases beyond bone health. This can reduce the risk of certain cancers, cardiovascular diseases, diabetes, and autoimmune disorders [66,208].

### 5.3. Reconciliation: The Need for Practical and Clinically Valuable Guidelines

The previous section reviews existing guidelines and highlights that these recommendations are predominantly based on the US IoM report [56], which primarily focuses on skeletal needs. Additionally, even the Endocrine Society guidelines have not adequately considered the requirements for other body systems.

The recommendations, evaluated variables, and data are insufficient for formulating country-specific guidelines aimed at public health—specifically for disease prevention and healthcare cost reduction. This section will address the broader issues surrounding evaluating vitamin D data relevant to these guidelines. It will explore determining the appropriate vitamin D intake and sun exposure necessary for optimal health.

### 5.4. Explanations for Differences Among Various Guidelines

Considerable confusion has arisen following the IoM [56] and European guidelines, as these are primarily based on bone-centric variables and have overlooked the broader pleiotropic benefits of vitamin D [53]. The recommended serum 25(OH)D concentrations of 20 ng/mL (50 nmol/L) and daily vitamin D doses of 400 to 800 IU are sufficient for musculoskeletal effects [27,29,50,152,209]. However, they are grossly inadequate for the extra-musculoskeletal benefits of vitamin D. As a result, many organizations have mistakenly interpreted the public health recommendations from the IoM as applicable to individual patients.

The broader and more comprehensive guidelines consider the pleiotropic effects of vitamin D and recommend a minimum serum 25(OH)D concentration of 30 ng/mL (75 nmol/L), which is currently accepted by most countries [52,53,69,94,210]. Depending on factors such as age and body size, oral vitamin D supplements typically range between 1000 and 4000 IU/day, with an average of about 2000 IU/day for adults. High-risk individuals—such as older adults, pregnant and lactating women [101,102], and those with a higher body mass index—may require higher doses, ranging from 4000 to 6000 IU/day [29,53]. These recommendations are tailored based on individual health conditions, geographic location, skin pigmentation, and other risk factors, including obesity.

### 5.5. Sun Exposure—Country-Specific Guidelines Needed

Although safe exposure to sunlight is the best and most natural source for increasing serum 25(OH)D concentration [16], millions of people find it challenging to achieve vitamin D sufficiency through sun exposure alone [211]. This challenge exists regardless of geographic latitude or whether individuals live in urban or rural areas, as most people spend over 95% of their time indoors, limiting their sun exposure. Additionally, conflicting public health recommendations from dermatologists regarding sun exposure and variations in geographic location and culturally specific habits make it nearly impossible to establish universal sun exposure guidelines that can serve as a reliable public health measure for ensuring adequate vitamin D intake [212,213].

Safe sun exposure should include protecting the face and eyes by wearing sunglasses and a brimmed hat to shield them from direct sunlight. The most effective time for the skin to generate vitamin D is between 10:30 a.m. and 1:30 p.m. when UVB rays from the sun reach an optimal (Zenith) angle for skin penetration. However, sunbathing indoors through glass windows is ineffective, as ultraviolet-B (UVB) rays are filtered out [211]. It is advisable to be exposed to the sun for about 30 min (or intermittently in brief periods) before applying sun-protection cream if staying in the sun longer.

For the majority, the only effective measure to prevent vitamin D deficiency is through vitamin D supplementation. Local recommendations for daily supplementation should consider native dietary and cultural habits, age, body mass index (or abdominal girth), disease burden, and clinically relevant health outcomes. Governments must address this widespread issue by promoting food fortification, as demonstrated by Finland and a few other countries, especially by targeting high-risk groups [214]. The overarching goal should be maintaining year-round serum 25(OH)D concentrations between 30 and 60 ng/mL (75 and 150 nmol/L).

### 5.6. Practical Guidance

Although measuring serum 25(OH)D concentration is the only reliable method to determine vitamin D status, excessive and inappropriate testing should be avoided to preserve resources and effectively manage costs. Using automated HPLC/MS/MS methods in bulk analysis, the total cost for automated 25(OH)D measurements is under USD 2 per test, including overhead. Despite this, many laboratories in the USA charge between USD 50 and USD 225 per test. Clinically applicable, on-the-spot finger-stick testing (or blood-spot mail-in testing) is becoming available with reasonable accuracy for clinical practice, which costs under USD 20/test. While the accuracy may be lower than that of the gold-standard LC/MS/MS method [215,216], it could still help more cost-effectively identify individuals with vitamin D deficiency. The barrier to using 25(OH)D assay is the cost and unavailability in many parts of the world.

Once point-of-testing measurements and other rapid and economical 25(OH)D estimations are routinely available, preferably under USD 10 to 15 per test, they will become a routine tool for better clinical management of patients. Until then, considering the cost, 25(OH)D measurement requests should be limited to baseline (i.e., pretreatment assessment in the high-risk/vulnerable populations) and, if appropriate, 4 to 6 months after initiating a loading dose plus maintenance vitamin D supplements to ensure concentrations are sustained at the desired therapeutic levels [51,217]. Routine testing of the population at large or those who are well and not at risk is unnecessary and inappropriate, except during screening for RCT recruitment.

### 5.7. Design and Interpretation Issues in RCTs for Nutrients, Specifically Vitamin D

RCTs have been uplifted as the gold standard for pharmaceutical drug trials and, by extension, have been applied to nutrient trials [218,219]. However, using RCTs in the context of nutrient trials is inappropriate and has led to failures. This is because the methodology that works for pharmaceuticals does not always translate effectively to nutrient studies. Dr. Robert Heaney, puzzled by the failures of vitamin D RCTs despite promising observational data [104,105], wrote an insightful paper on how to reconcile these deviations and how RCTs for vitamin D should be appropriately conducted [19]. One recommended approach is to identify a significant vitamin D-deficient population and supplement only the treatment group with vitamin D. Such studies have primarily been conducted in pregnant populations outside the USA [220,221,222].

The results of RCTs focusing on vitamin D supplementation during pregnancy have demonstrated significant improvements in various birth outcomes. However, conducting such studies in the USA is ethically challenging, making them difficult to replicate. Dr. Heaney emphasized that baseline 25(OH)D levels should be known for each participant in such trials and accounted for in the final analysis to avoid confounding factors that could invalidate the study [19]. For example, if a pharmaceutical company were conducting an RCT on a statin drug to lower cholesterol, it would never include participants already taking another statin in the control group, as this would likely render the study ineffective. Similarly, failing to account for existing vitamin D levels in RCT on vitamin D supplementation could lead to inconclusive or null results.

When conducting a vitamin D RCT, it is crucial to account for baseline 25(OH)D levels, as they can significantly modify the study outcomes. A better trial design would involve considering these baseline levels, particularly in the control group, and adjusting for them in the final statistical analysis. This approach is consistent with the intent-to-treat model used in drug RCTs, provided the analysis is pre-specified before the trial begins. Correcting baseline levels is vital for nutrient RCTs, as failing to do so can obscure the actual effects of supplementation. A recent paper by Weiss ST et al. (2024) [223] demonstrated the importance of this adjustment. The study highlighted the importance of accounting for initial 25(OH)D levels in trial subjects to ensure accurate results. This paper is critical reading for anyone involved in designing or reanalyzing vitamin D trials, as it provides evidence of how correcting for baseline 25(OH)D levels can fundamentally change the interpretation of trial data.

The Vitamin D Antenatal Asthma Reduction Trial (VDAART) is an excellent example of a well-designed and executed pregnancy intervention trial. However, it initially yielded disappointing results when analyzed using the standard intent-to-treat method [224,225]. However, the findings were remarkable when the authors reanalyzed the data, adjusting for baseline 25(OH)D levels. They revealed a statistically significant reduction in asthma incidence in children linked to maternal vitamin D supplementation during pregnancy.

This reanalysis demonstrated how correcting baseline vitamin D levels can reveal significant previously obscured effects. They also reexamined data from another trial conducted in Denmark, the COPSAC trial. Like VDAART, the original COPSAC trial did not find a significant impact of vitamin D supplementation on asthma prevention [226]. Nevertheless, once the data were reanalyzed with baseline 25(OH)D corrections, the results became highly significant, further validating the importance of this methodological adjustment in nutrient trials [226].

### 5.8. Importance of Reanalyzing Data Based on Baseline Serum 25(OH)D Concentrations

Vitamin D clinical research has been complicated by the difficulties in conducting adequately statistically powered and properly designed RCTs, which are typically regarded as the gold standard for evaluating scientific questions and approving new pharmaceuticals [218,219]. However, RCTs in the context of nutrient trials, like vitamin D, face unique challenges that can lead to erroneous conclusions. Issues such as study design flaws, bias, and improper data evaluation are common, making it difficult to draw accurate conclusions.

One notable example is the VITAL study [165], which has been criticized for its major design faults. Consequently, primary outcomes and interpretations from such studies should not be generalized or included in developing public health guidelines. Unlike pharmaceutical trials, nutrient trials require specific attention to factors such as baseline nutrient status, the need for cofactors, dose–response relationships, and population-specific needs. When these factors are not considered, the results, like those of the VITAL study [165], may fail to reflect the potential benefits of supplementation and should not be relied upon to inform guidelines.

Irrespective of the sample size, funding and inherent biases, including factors such as investigator bias, funding bias (e.g., from large pharmaceutical companies), or critical design errors, preclude the proper interpretation of data. This applies even to mega RCTs like the VITAL study, where significant design flaws invalidate their relevance in generating country- or region-specific vitamin D guidelines. Furthermore, sensationalized interpretations by the media can exacerbate the problem, as misleading information spreads rapidly, potentially leading to long-term harm. These flawed conclusions could misguide public health efforts, making it essential to critically evaluate the design and execution of RCTs before incorporating their findings into any guidelines.

Adequately powered RCTs that measure serum 25(OH)D concentrations in participants at baseline, establish and maintain a targeted serum 25(OH)D level throughout the study (rather than relying solely on dietary intakes), are conducted for sufficient durations to assess primary endpoints, and involve randomized subjects who are vitamin D deficient are associated with favorable clinical outcomes. Most of these studies indicate that achieving and maintaining serum 25(OH)D concentrations above 30 ng/mL reduces the risks of several diseases and decreases all-cause mortality [227,228,229].

### 5.9. Vitamin D Supplementation Protocols

Data suggest that daily vitamin D supplements are more physiological than infrequently administered higher doses. The effectiveness and benefits of vitamin D are markedly diminished when taken less frequently than once a month [52,53,230]. Such non-physiological dosing creates a discordance between tissue and serum levels, adversely affecting immune function. Total body vitamin D deficiency can be estimated in international units based on body weight (or fat mass) and baseline serum 25(OH)D concentration. By administering a calculated loading dose over a few weeks, followed by a maintenance dose, it is possible to avoid the need for repeated serum 25(OH)D measurements, which may be unaffordable for many [7].

More than 95% of individuals prescribed a high loading dose of vitamin D attain target level of serum 25(OH)D concentration. Those cases typically involve gastrointestinal absorption abnormalities, medications that increase vitamin D catabolism, or obesity. Individuals in this category may require a repetition of a similar loading regimen or a higher dose to achieve a serum 25(OH)D concentration greater than 30 ng/mL. Additionally, they will likely need a higher daily maintenance dose of vitamin D. Meanwhile, public education on the benefits of sensible sun exposure and consuming vitamin D-fortified foods and supplements can help achieve the desired vitamin D status in over 99% of the population [122,231]. Selective food fortification programs for nutrients are cost-effective and successfully implemented worldwide, including programs for iodine, iron, and vitamins A and D [232,233,234].

Virtually everyone can maintain sufficient vitamin D by combining a healthy diet and physical exercise to reduce the burden of obesity, along with daily sun exposure or the consumption of vitamin D supplements. This public health approach is highly cost-effective in maintaining health and reducing healthcare costs [235]. This strategy should include increasing the availability of high-quality and affordable vitamin D supplements in countries where food fortification is not feasible [117]. Additionally, maintaining vitamin D status and metabolic functions, such as intracellular enzymatic reactions and mitochondrial respiration, is enhanced by adequate availability of cofactors, including omega-3 fatty acids, B vitamins, magnesium, vitamin K, selenium, and zinc [126].

## 6. Discussion

Vitamin D deficiency is the most prevalent nutritional deficiency across all age groups worldwide. This is primarily due to reduced sun exposure (sun-avoiding behavior), limited vitamin D-rich foods, misleading guidelines, and a lack of consensus on supplementation. Infants require at least 400 IU of vitamin D in their first year, while younger children need over 1000 IU daily, with requirements increasing as they grow. Most healthy adults need a minimum of 4000 IU daily, and older healthy adults generally require at least 6000 IU daily. Vitamin D_3_ is the preferred form of supplementation. After ingesting vitamin D_3_, changes in serum 25(OH)D concentration can typically be observed within three days.

Vitamin D is essential for health and survival, yet optimal supplementation and management guidelines can vary significantly due to geographic, dietary, and health factors. There is a pressing need to develop region- or country-specific vitamin D guidelines rather than relying solely on Western recommendations. Understanding vitamin D biology and physiology and critically evaluating errors in existing guidelines (particularly flawed concepts) would allow the creation of tailored recommendations. Using such, countries can better meet their unique needs and improve public health outcomes related to vitamin D. Individuals with obesity, gastrointestinal absorption issues, or conditions that impair vitamin D synthesis or increase their catabolism may require considerably higher doses than standard recommendations to achieve optimal health.

Contraindications for vitamin D supplementation, or its administration with caution, include individuals who have had or may develop hypercalcemia. This group encompasses but is not limited to those with granulomatous diseases (e.g., tuberculosis and sarcoidosis) and genetic conditions like Williams syndrome. Circulatory and body stores of D_3_ and 25(OH)D can be replenished through safe sun exposure and/or vitamin D_3_ (cholecalciferol) supplementation. For individuals with vitamin D deficiency, initial loading doses can rapidly restore serum and body stores. A typical regimen involves administering 50,000 IU two to three times per week for several weeks or a one-time dose of 200,000 IU, ideally following a fatty meal to enhance absorption. The required dose varies depending on baseline serum 25(OH)D levels [51,62,188]. Any loading dose must be followed with a suitable daily dose of vitamin D.

Considering the half-life, storage capacity, and time required to reach equilibrium, the optimal timing for rechecking serum 25(OH)D levels is approximately 10 to 14 weeks after completing a loading dose. If the level remains deficient, a second course of cholecalciferol (D3) may be prescribed for 4 to 6 weeks. The goal should be to achieve a minimum serum 25(OH)D concentration of 30 ng/mL. However, to address most disorders (see Figure 2), it is recommended to maintain serum 25(OH)D concentrations above 50 ng/mL within the range of 40 to 80 ng/mL. Unless regularly exposed to sunlight, most will require a daily maintenance dose of vitamin D_3_ supplements for better health, especially those in temperate climates. Given ethnic, cultural, dietary, social, and geographic differences, each country should develop its own vitamin D and calcium supplementation guidelines.

## 7. Conclusions

Until now, vitamin D requirements have been based exclusively on its role in calcium metabolism and endocrine function. This faulty assumption should be changed. The calcium metabolic system requires relatively low amounts of vitamin D compared to these other systems. This narrow focus has significantly underestimated the proper requirements, which must also encompass the needs of genomic and intracrine/paracrine systems. This broader consideration must be applies to both dietary supplementation and the circulating levels of 25(OH)D concentrations. Reliance solely on skeletal benefits (e.g., preventing rickets in children and osteomalacia in adults and maintaining physiological circulatory PTH levels) for recommending vitamin D intake and 25(OH)D blood levels is flawed and must be abandoned. This mistake has led to misguided and inadequate guidelines, ultimately misinforming the public and administrators harming the public. This review demonstrates how the distinct needs of various body systems can be met with proper vitamin D intake to address differential tissue requirements to overcome all common disorders, as shown in Figure 2. It also offers insight into the importance and practical guidance on developing new country- and region-specific vitamin D clinical guidelines.

## Figures and Tables

**Figure 1 nutrients-16-03969-f001:**
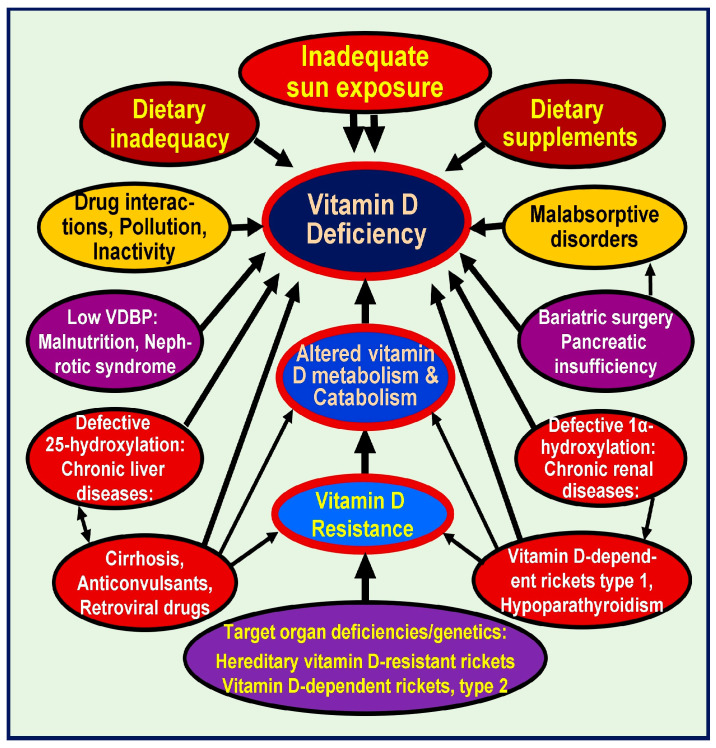
Three sources of vitamin D and groups of causes that lead to vitamin D deficiency, including altered gastrointestinal absorption, enhanced catabolism, and vitamin D resistance syndromes.

**Figure 2 nutrients-16-03969-f002:**
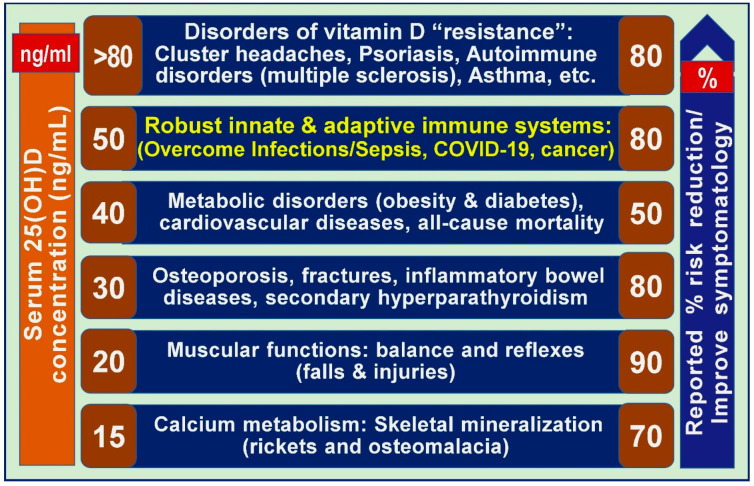
Varying serum 25(OH)D concentrations are needed to counteract different diseases. The average serum 25(OH)D concentrations needed to prevent or reduce the risks of a few common diseases are illustrated in each column [modified from Wimalawansa, S.J., 2023] [4].

**Figure 3 nutrients-16-03969-f003:**
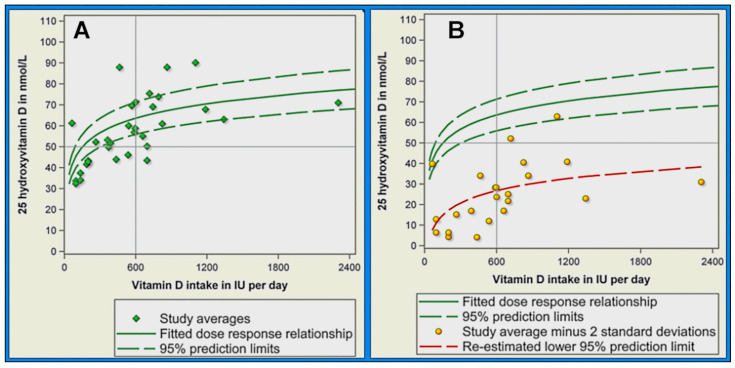
(**A**). Dose–response relationship of vitamin D intake and serum 25 hydroxyvitamin D (fitted dose–response relationships). (**B**) Dose–response relationship of vitamin D intake and serum 25 hydroxyvitamin D after the 95th prediction limit re-estimations. (after Veugelers and Ekvaru, 2014) [156].

**Figure 4 nutrients-16-03969-f004:**
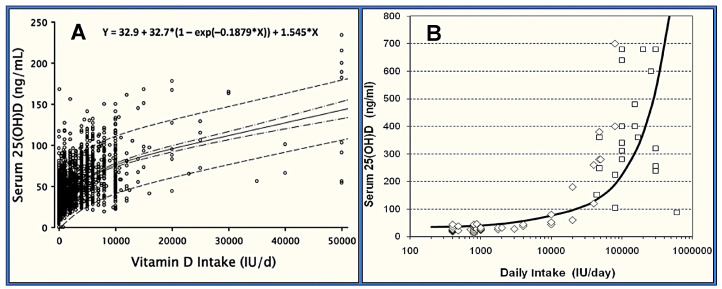
(**A**) Serum 25(OH)D as a function of daily vitamin D intake, the best-fit line with the confidence limits, and the 95% probability (two outer dashed lines) for the entire cohort [92]. (**B**) The regression line from Figure (**A**) using published high-dose vitamin D data [92] (the *Y*-axis is logarithmic). The means of controlled dosing studies (n = 48) are illustrated as diamond-shaped symbols. Individual values from reported vitamin D intoxication cases (n = 21) are depicted by square symbols (after Heaney, RP, 2011 [158]).

**Table 1 nutrients-16-03969-t001:** Vitamin D supplementation guidelines for the Arab population [77] ^#^.

	Category of Persons	Supplementation Guidelines
A	Premature infants	400 to 800 IU/day, starting from the first days of life
B	Breastfed infants	400 IU/day to age six months and 600 IU/day between 6 and 12 months, depending on daily intake of total vitamin D and sun exposure
C	Children and adolescents, 1 to 18 yrs	600 to 1000 IU/day, depending on body weight
D	Adults older than 18 years	1000 to 2000 IU/day throughout the year
E	Elderly (males and females older than 65 years)	2000 IU/day throughout the year
F	Pregnant and breastfeeding women	2000 IU/day from the first trimester of pregnancy *
G	Obese individuals and those with metabolic syndrome	Supplementation at 2000 IU/day (50 μg/day) throughout the year, depending on body weight
H	Individuals with dark skin and night workers	2000 IU/day (50 μg/day) throughout the year, depending on body weight

^#^ These guidelines are also outdated now and based on older studies. * The current global recommendation for vitamin D for pregnancy and lactation is between 4000 and 6000 IU [103].

**Table 2 nutrients-16-03969-t002:** The published country-specific guidelines for vitamin D to control chronic diseases.

Country	Agency	Recommended Sufficiency Serum Levels	Recommended Doses (IU/Day)	Focus on Chronic Diseases	References
United States	Institute of Medicine (IOM) The Endocrine Society (TES)	Serum 25(OH)D levels are 20 ng/mL (50 nmol/L) for the general population, according to IoM. The Endocrine Society recommends at least 30 ng/mL (75 nmol/L).	IoM: adults: 600–800 IU/day, with higher doses for those at risk of deficiency. TES: chronic conditions, higher intakes (1500–2000 IU daily) for optimal health and disease prevention (2011) [18,57] —revert to 600 IU doses suggested by the IoM in 2024 recommendations [32].	Strong focus on bone health (osteoporosis), with emerging evidence and recommendations for cancer, cardiovascular diseases, and autoimmune conditions. However, no changes were introduced despite the overwhelming presence of published scientific evidence.	[56] [50]
	American Academy of Developmental Medicine and Dentistry (AADMD)	Individuals with intellectual and developmental disability (IDD) are recommended to maintain serum 25(OH)D above 30 ng/mL.	For IDD, doses between 2000 and 5000 IU/day are recommended. Alternatively, regimens of 15,000 to 30,000 IU per week or 50,000 IU twice a month [51,187,188].		[57]
United Kingdom	National Institute for Health and Care Excellence (NICE)	25(OH)D levels of 25 nmol/L or above are sufficient.	400 IU/day for the general population, focusing on supplementation during autumn and winter.	It primarily focused on bone health and prevention of rickets and osteomalacia, with cautionary advice (i.e., not to use supplementation for applications beyond bone health).	[200]
Canada	Osteoporosis Canada, Health Canada	25(OH)D levels of 30 ng/mL (75 nmol/L) are recommended for bone health.	400–1000 IU/day for adults, with higher doses (2000 IU/day) for at-risk people.	Focus on osteoporosis prevention; emerging guidelines consider roles in cancer and cardiovascular health. However, no dose changes are recommended.	[201]
Australia and New Zealand	Australian and New Zealand Bone and Mineral Society (ANZBMS)	25(OH)D levels of 50 nmol/L or more considered sufficient.	Depending on age, 400–800 IU/day, with higher recommendations during winter or for at-risk people.	It was primarily focused on musculoskeletal health, with some recommendations for broader chronic disease prevention.	[202]
India	Indian Council of Medical Research (ICMR)	25(OH)D levels of 20–30 ng/mL (50–75 nmol/L) are sufficient.	400–1000 IU/day, with higher doses recommended for those with limited sunlight exposure.	Focus on bone health, with increasing attention to diabetes and cardiovascular diseases.	[203]
Nordic Countries	Nordic Council of Ministers	50 nmol/L is the threshold for sufficiency.	400–800 IU/day for the general population, with adjustments based on seasonal sunlight availability.	Focus on bone health, with guidelines also considering the prevention of multiple sclerosis and other autoimmune diseases.	[204]
Middle East and North Africa (MENA)	Various national health ministries and organizations	20–30 ng/mL (50–75 nmol/L) considered sufficient.	Doses of 1000–2000 IU/day are recommended due to prevalent deficiency.	Emphasis on bone health and emerging concerns about diabetes, cardiovascular diseases, and immune function.	[205,206]
Arab population	Persian Gulf-region guidelines	Minimum of 30 ng/mL	800 to 2000 IU/day based on age and conditions.	The recommendation was mainly for bone health, but also for pregnancy, lactation, and dark-skinned persons: 2000 IU/day.	[77] [207]
Polish	Polish vitamin D guidelines		The report also recommended a one-size-fits-all principle: to take 4000 IU/day or 7000 IU/week for ages between 1 and 90 years, which makes little sense.	The report recommends using calcifediol, which is 15 to 20 times more expensive than cholecalciferol and is likely to cause more adverse effects [7].	[53]
Japan	Japanese Society for Bone and Mineral Research (JSBMR)	20 ng/mL (50 nmol/L) or more recommended.	Depending on age, 400–800 IU/day, with seasonal adjustments.	Focus primarily on bone health, with an emerging interest in its role in cancer prevention.	[172]

## Data Availability

Data included in the article are referenced in the article.

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
