# Peer review of "Integrating Endocrine, Genomic, and Extra-Skeletal Benefits of Vitamin D into National and Regional Clinical Guidelines"

_nutrients, 2024, doi:10.3390/nu16223969_

Round 1

Reviewer 1 Report

Comments and Suggestions for Authors

This work is a broad review of the current literature regarding vitamin status and the recommendations of intake/supplementation. Authors provide a general overview of main aspects and considerations of vitamin D and its metabolism and effects. However, some improvements could be made:

-          Tittle: I think the tittle does not reflect exactly the content of this work; this is not exactly the development of guidelines, is more a compilation of the current evidence.

-          Line 12: the “T” of the abbreviation RCTs corresponds to “trials”, please, change it.

-          Line 21 and 23: However is used twice consecutively; Maybe it would be more appropriate the use of “additionally/moreover” in line 23.

-          Lines 21-27: I suggest rewriting these sentences to highlight the need of these guidelines’ creation in a clearer way.

-          Line 31: similar comment with the term “country-specific”. This is an international journal, and it must be clear which country is for.

-          First paragraphs of introduction section: references are lacking for the statements made.

-          Line 79: the introduction of this last paragraph about breast milk is abrupt, some introductory sentence would be convenient.

-          Line 88: “it hydroxylated” possible spelling mistake.

-          Lines 88-93: references are lacking.

-          Line 98: 1,25(OH)2D, number 2 should be such a subscript (check this point throughout the manuscript)

-          Line 101: “below ten ng/mL” would be more appropriate use the number 10

-          Line 110: “The IoM and the latest TES guidelines”, please indicate the full name of the abbreviations the first time they appear in the text.

-          Lines 182-193: It would be interesting to indicate the equivalence between IU and ng/mL

-          Line 202: Please use number to express concentrations (10 ng/mL), follow this criteria throughout the text.

-          Figure 1 is informative, but the appearance should be improved.

-          The full name and abbreviation of the Institute of Medicine (IOM) appears in line 405 when you have previously used in the text. Please, correct this point.

-          Lines 492-497:  the equivalence among different units should be indicated to improve the understanding (ng/mL; IU; nmol/L)

-          If figures 3 and 4 have been extracted from other articles, they should have author’s permission to use them.

-          I miss some table or figure summarizing the main aspects of this manuscript. The content is good but long. For readers, it would be interesting to summarize more important aspects regarding the objective of this manuscript.

-          Point 6 seems more like a conclusion than a discussion section. Maybe it would be appropriate to summarize a little bit this part and use it as conclusion.

Author Response

(Old Title)
Developing National Guidelines for Vitamin D Intake

New Title:
The Importance of Capturing Extra-Skeletal Benefits in Developing National
Guidelines for Vitamin D Intake

Please find the author’s response to each query by three referees. 

RESPONSES ARE IN BOLD LETTERS.

The authors are greatly appreciated for the thoughtful suggestions.

---------------------------------------------------------

Reviewer #1:
This work is a broad review of the current literature regarding vitamin status and the recommendations of intake/supplementation. Authors provide a general overview of main aspects and considerations of vitamin D and its metabolism and effects. However, some improvements could be made:

- Tittle: I think the tittle does not reflect exactly the content of this work; this is not exactly the development of guidelines, is more a compilation of the current evidence.

Response: We agree and have changed the title to Inclusion of extra-skeletal benefits During Developing National Guidelines for Vitamin D Intake. We hope this is suffice.

- Line 12: the “T” of the abbreviation RCTs corresponds to “trials”, please change it.

Done:  Thank you: Add parenthesis to ” Clinical: for clarity.

- Line 21 and 23: However is used twice consecutively; Maybe it would be more appropriate the use of “additionally/moreover” in line 23.

Thank you: This was corrected. Replaced with Additionally, as suggested.

- Lines 21-27: I suggest rewriting these sentences to highlight the need of these guidelines’ creation in a clearer way.

Thank you: We have added a few sentences to highlight this important point.

-  Line 31: similar comment with the term “country-specific”. This is an international journal, and it must be clear which country it is for.

The Authors have modified it as suggested.

- First paragraphs of introduction section: references are lacking for the statements made.

Thank you: The authors have added supporting references in the first paragraphs to justify this sentence. I have added the requested refs.

- Line 79: the introduction of this last paragraph about breast milk is abrupt, some introductory sentence would be convenient.

Thank yoy: A The authoirs have addressed this issue.

- Line 88: “it hydroxylated” is a possible spelling mistake.

This is now corrected.          

- Lines 88-93: references are lacking.

The authors have added references to compensate what was missing.

-  Line 98: 1,25(OH)2D, number 2 should be such a subscript (check this point throughout the manuscript)

Corrected though the manuscript.

- Line 101: “below ten ng/mL” would be more appropriate use the number 10

Switched ten into 10 as suggested.

- Line 110: “The IoM and the latest TES guidelines”, please indicate the full name of the abbreviations the first time they appear in the text.

Thak you again: We have c added the full name for both.

- Lines 182-193: It would be interesting to indicate the equivalence between IU and ng/mL

A couple of sentences were added to clarify the weight to IU relationship.

- Line 202: Please use number to express concentrations (10 ng/mL), follow this criteria throughout the text.

We have modified this now.

- Figure 1 is informative, but the appearance should be improved.

We have made slight adjustments to the Figure However, we are not clear what precisely the way the reviewer would  like us to improve the figure. 

-  The full name and abbreviation of the Institute of Medicine (IOM) when you have previously used in the text. Please, correct this point.

This is now accomplished – Thank you.

- Lines 492-497: the equivalence among different units should be indicated to improve the understanding (ng/mL; IU; nmol/L).

We have modified the sentences to make them clearer and more meaningful.

  • If figures 3 and 4 have been extracted from other articles, they should have author’s permission to use them.
  • After Veugelers and Ekvaru, 2014) [159].
       AND
    After Heaney, RP, 2011, [162]).

- Reviewer-- I miss some table or figure summarizing the main aspects of this manuscript. The content is good but long. For readers, it would be interesting to summarize more important aspects regarding the objective of this manuscript.

We are not exactly sure what was meant by the reviewer. 

- Point 6 seems more like a conclusion than a discussion section. Maybe it would be appropriate to summarize a little bit this part and use it as conclusion.

Thank you:  we have modified the discussions and created specific Concluding remarks at the end.

Greatly appreciate your constructive suggestions.

Reviewer 2 Report

Comments and Suggestions for Authors

Very well-written paper. 

I suggest to include as ection or some case studies/reports or examples of successful national guidelines to illustrate practical applications. E.g. Canada/Finland/Saudi Arabia fortifications and maybe Australia sun for different types of skin colors. https://www.theguardian.com/australia-news/2024/feb/13/australias-sun-safety-guidelines-updated-to-take-account-of-diverse-skin-types

Author Response

(Old Title) Developing National Guidelines for Vitamin D Intake

New Title:
The Importance of Capturing Extra-Skeletal Benefits in Developing National
Guidelines for Vitamin D Intake

RESPONSES ARE IN BOLD LETTERS.

Please find the author’s response to each query by three referees.
The authors are greatly appreciated for the thoughtful suggestions.

-------------------------------------------------

Referee #2:

Very well-written paper. 

I suggest to include as section or some case studies/reports or examples of successful national guidelines to illustrate practical applications. E.g. Canada/Finland/Saudi Arabia fortifications and maybe Australia sun for different types of skin colors. The authors respectfully state cause reports, and adding sections of skin color etc, might be out of scope for this paper. Considering the high word counts, adding care reports, albeit brief, will significantly increase the length of the manuscript.

Thank you for the suggestion. However, adding cause reports and skin color-related paragraphs or sections would significantly increase the length of the manuscript.

Considering it already has a high word count and the content seems to be out of the scope of the manuscript.

https://www.theguardian.com/australia-news/2024/feb/13/australias-sun-safety-guidelines-updated-to-take-account-of-diverse-skin-types

Thank you for  your constructive suggestions.

Reviewer 3 Report

Comments and Suggestions for Authors

This is a very meaningful review that elaborates on the function of vitamin D, blood levels, and the current research progress on this subject. A detailed analysis of the guidelines for VD supplementation from the two major U.S. organizations, the IOM and the TES, and vitamin D clinical practice guidelines from several other countries, aims to fill the gaps in current vitamin D clinical practice guidelines and suggests a framework for the development of more effective and targeted recommendations to prevent disease and improve public health. I applaud the authors for such a comprehensive summary. I have only a few suggestions for this review.

1.      There is ample evidence to suggest that vitamin D plays an important role in the impact on bone and exoskeleton health. However, the dosage of supplements and the ideal level they should achieve are highly controversial. This is related to multiple factors, including but not limited to diet, supplements, personal health status, outdoor activity time, and genotype, making it difficult to unify. I suggest these factors can be included in the discussion.

2.      A large part of this article is devoted to the fact that the recommended dose of vitamin D is too low to achieve the desired level, but at the same time we should also consider the question of the safe dose of vitamin D supplementation, and in this way we should also take into account the upper limit of the maximum dose of intake for different populations (tolerable maximum intake).

Author Response

(Old Title) Developing National Guidelines for Vitamin D Intake

(New Title):
The Importance of Capturing Extra-Skeletal Benefits in Developing National
Guidelines for Vitamin D Intake

RESPONSES ARE IN BOLD LETTERS.

Please find the author’s response to each query by three referees.
The authors are greatly appreciated for the thoughtful suggestions.

Reviewere #3:

This is a very meaningful review that elaborates on the function of vitamin D, blood levels, and the current research progress on this subject. A detailed analysis of the guidelines for VD supplementation from the two major U.S. organizations, the IOM and the TES, and vitamin D clinical practice guidelines from several other countries, aims to fill the gaps in current vitamin D clinical practice guidelines and suggests a framework for the development of more effective and targeted recommendations to prevent disease and improve public health. I applaud the authors for such a comprehensive summary. I have only a few suggestions for this review.

  1. There is ample evidence to suggest that vitamin D plays an important role in the impact on bone and exoskeleton health. However, the dosage of supplements and the ideal level they should achieve are highly controversial. This is related to multiple factors, including but not limited to diet, supplements, personal health status, outdoor activity time, and genotype, making it difficult to unify. I suggest these factors can be included in the discussion.

The authors have added the following:
Age, BMI, diet, sun exposure, supplement dosage, personal health status, and genotype all contribute to the variability in serum levels of vitamin D, making it essential to personalize dosing of vitamin D. See line 247

And we expanded the section as suggested by the referee.

  1. A large part of this article is devoted to the fact that the recommended dose of vitamin D is too low to achieve the desired level, but at the same time we should also consider the question of the safe dose of vitamin D supplementation, and in this way we should also take into account the upper limit of the maximum dose of intake for different populations (tolerable maximum intake).

Thank you: The authors have added brief sections on Vitamin D-related adverse effects at the end of the Section 2.4 as suggested by the refree.

Greatly appreciate your constructive suggestions.